# Exploring evolution-aware & -free protein language models as protein function predictors

**Mingyang Hu**[*]
Westlake University
humingyang@westlake.edu.cn

**Fajie Yuan**[*†]
Westlake University
yuanfajie@westlake.edu.cn

**Kevin K. Yang**
Microsoft Research New England
yang.kevin@microsoft.com

**Fusong Ju**
Microsoft Research Asia
fusongju@microsoft.com

**Jin Su**
Westlake University
sujin@westlake.edu.cn

**Hui Wang**
Westlake University
wanghui@westlake.edu.cn

**Fei Yang**
Zhejiang Lab
yangf@zhejianglab.com

**Qiuyang Ding**
Westlake University
dingqiuyang@westlake.edu.cn

## Abstract

Large-scale Protein Language Models (PLMs) have improved performance in protein prediction tasks, ranging from 3D structure prediction to various function predictions. In particular, AlphaFold [23], a ground-breaking AI system, could potentially reshape structural biology. However, the utility of the PLM module in AlphaFold, Evoformer, has not been explored beyond structure prediction. In this paper, we investigate the representation ability of three popular PLMs: ESM-1b (single sequence) [35], MSA-Transformer (multiple sequence alignment) [30] and Evoformer (structural), with a special focus on Evoformer. Specifically, we aim to answer the following key questions: (i) Does the Evoformer trained as part of AlphaFold produce representations amenable to predicting protein function? (ii) If yes, can Evoformer replace ESM-1b and MSA-Transformer? (iii) How much do these PLMs rely on evolution-related protein data? In this regard, are they complementary to each other? We compare these models by empirical study along with new insights and conclusions. All code and datasets for reproducibility are available at https://github.com/elttaes/Revisiting-PLMs.

## 1 Introduction

Proteins perform the majority of biological activities. It is, therefore, crucial to decipher the mechanisms underlying their structural and functional properties. The canonical *sequence-structure-function* relationship enables the success of sequence-based machine learning methods that infer protein structure and function from amino acid (AA) sequence. Large-scale protein language models (PLMs) with self-supervised pretraining on tens of millions to billions of proteins (PLMs) [28, 35, 13, 5] are the current state-of-the-art in predicting function and fitness from sequences.

Meanwhile, AlphaFold [23], trained on experimental 3D protein structures from the Protein Data Bank (PDB) [39] can approach the resolution of experimental structures for most protein sequences.

---

[*]Equal Contribution

[†]Corresponding author. Fajie designed the idea and led the research. Mingyang performed the research and led all experiments. Kevin provided important guidance for this research and worked on a part of paper writing. Jin performed experiments for the MSA generation.

36th Conference on Neural Information Processing Systems (NeurIPS 2022).

Its multiple sequence alignment representation module, Evoformer, combines new deep learning machinery, a PLM residue reconstruction task, and structural supervision in the form of a distogram. Like MSA-Transformer [30], Evoformer takes a family of evolutionarily-related and aligned protein sequences as input, in contrast to PLMs such as ESM-1b [35] and TAPE [28], which only take individual protein sequences. Thus, for short, we refer to the former two models as evolution-aware PLMs and the latter two as evolution-free PLMs.

Despite the remarkable success of AlphaFold in predicting structure from sequence, it is unknown whether its Evoformer module can be applied to other problems, in particular predicting protein function and fitness. Deciphering AlphaFold rather than treating it as a black box is beneficial to both AI and biology communities. Therefore, we attempt to answer the following key questions.

**Q(i): Does the Evoformer trained during AlphaFold training learn general-purpose protein representations that can be used for various function prediction tasks?** Unlike ESM-1b and MSA-Transformer, Evoformer is trained with supervision from 3D structures. In addition, the second part of AlphaFold, the Structure Module, built on top of the 48 Evoformer blocks, is much more complex and deeper than the traditional (linear) classification head used in ESM-1b and MSA-Transformer. These differences make the function representation ability of Evoformer an open question.

**Q(ii): If Evoformer's representation is general, does it outperform ESM-1b and MSA-Transformer on downstream tasks?** While these three models are trained with different parameter sizes and datasets (see Table 1), we believe the comparison results are still valuable because they are currently the most advanced PLMs. Training these large models from scratch is out of the reach of most academic research groups due to the computation and cost involved. In addition, we also investigate the utility of MSA-Transformer on function prediction tasks.

**Q(iii): How much does the performance of evolution-aware PLMs rely on the input MSAs? Can evolution-free PLMs assist evolution-aware PLMs in terms of MSA construction?** Both Evoformer and MSA-Transformer take sets of aligned sequences (MSAs) as input. We investigate the effect of MSA quality and depth on function prediction. On the other side, ESM-1b can simply formulate MSA construction as the remote homology detection task. Then, it is interesting to know whether ESM-1b-constructed MSAs can be used as inputs to Evoformer and MSA-Transformer.

We address the above questions through comprehensive empirical studies on a variety of structure and function prediction tasks. Note although AlphaFold (i.e. Evoformer + Structure Module) can accurately predict protein 3D structures, we investigate the ability of Evoformer to perform other structure prediction tasks, such as secondary structure prediction and contact map prediction. We make the following key observations:

(i) The AlphaFold-trained Evoformer produces representations that are useful for both structure and function prediction, as shown on two structure prediction tasks, two function annotation tasks [48], and two fitness score prediction tasks [28].

(ii) Evoformer representations are useful for both protein-level and residue-level prediction tasks.

(iii) Evoformer is superior to ESM-1b and MSA-Transformer for structure prediction and novel miniprotein stability prediction, but inferior to ESM-1b on other function prediction tasks. It performs poorly for zero-shot fitness prediction tasks [27] compared to ESM-1b and MSA-Transformer.

(iv) Evolution-aware PLMs are superior to evolution-free ESM-1b model only in the structure prediction tasks, but in general, are worse than ESM-1b in most function prediction tasks.

(v) Like structure prediction, evolution-aware PLMs are also sensitive to the amount of MSAs when predicting protein functions. In addition, their performance using ESM-1b-constructed MSAs as input is comparable to the performance using MSAs generated by Jackhmmer [22] or HHblits [32].

## 2   Related Work

**Protein language models**   The volume of protein data has exploded over the last decade with the advancement of new DNA sequencing technologies. Early work learned protein representations with LSTMs [18, 4, 1]. Recently, with the advent of large transformer models in natural language processing (NLP), large PLMs using the Transformer [43] architecture and BERT [12] denoising

Table 1: Model descriptions. 'Para.', 'M', & 'seqs' denotes parameters, million, & protein sequences.

| Model | Embedding | Layers | Para. | Training database |
|---|---|---|---|---|
| ESM-1b | 1280 | 33 | 650M | UniRef50 (27M seqs) |
| MSA-Transformer | 768 | 12 | 100M | UniRef50 (26M MSAs) |
| Evoformer (No Template) | 256 & 128 | 48 | 88M | PDB (190K structures + MSAs) |

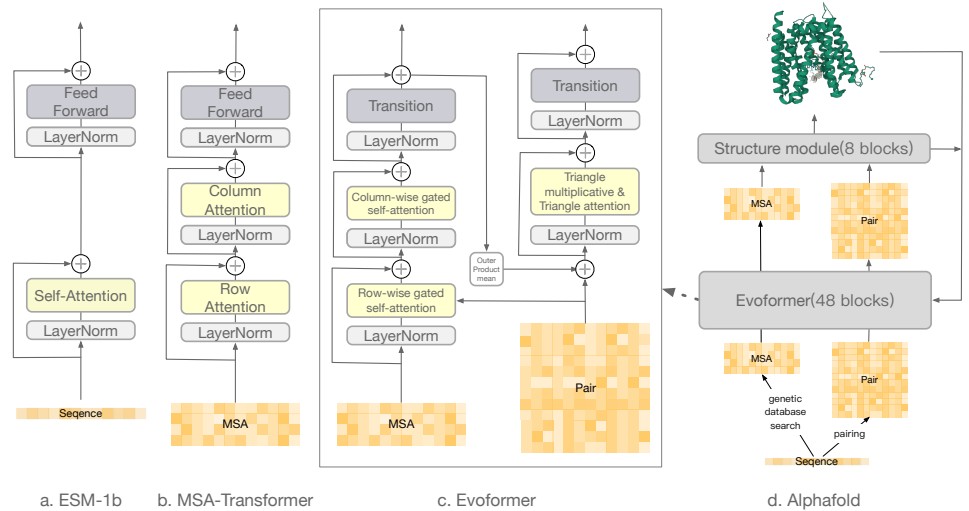

Figure 1: Core modules of the three PLMs.

task have been widely adopted [35, 28, 13].* PLMs trained on large sequence databases have been successfully applied for various protein related tasks, including secondary structure prediction [28, 18, 35, 13], contact prediction [29, 35], 3D structure prediction [23, 2], annotation prediction [7], signal peptide prediction [40], intracellular localization prediction [41], protein-protein interaction prediction [14], and fitness prediction [1, 10, 19, 20, 27]. ESM-1b [35] found that residue-residue contacts can be recovered from the learned representation, identifying the close relationship between Transformer attention and biological features. Following this, [44] and [29] further studied the interpretability of the attention map as contact map. ProtTrans [13] benchmarked a variety of BERT-like PLMs, including TAPE, ESM-1b, ProtTrans, and MSA-Transformer. Since ProtTrans with the BERT architecture does not exceed ESM-1b, we choose to compare ESM-1b, MSA-Transformer, and Evoformer for this study. Table 1 and Figure 1 provide more details about the three models.

**Structure, function, and fitness prediction** A protein's primary structure (i.e. AA sequence) determines its 3D structure, which in turn determines its functional properties. This relationship underlies the success of PLMs, which infer protein structure and function from the raw sequence. Secondary structure (SS) prediction [38] is an easier task than predicting the 2D contact map [45] and 3D structure [23]. MSA-Transformer and ESM-1b can make accurate unsupervised contact predictions, which are an important input for the 3D structure generation.

While remarkable progress has been made in structure prediction, it is unknown whether this AlphaFold-triggered revolution transfers to other tasks, in particular function prediction. Protein function is a broad term that refers to any biological or biochemical roles in organisms. In this paper, we focus on functional annotation prediction (classification task) [48, 6] and fitness prediction (including both regression [10, 28] and zero-shot prediction [19, 27] tasks), as shown in Figure 2. Compared with annotation prediction, fitness prediction is more challenging since (1) there are routinely few or no laboratory labels for supervision; (2) protein AA sequences are highly similar given the same wild-type protein.

---

*Note that the bioRxiv version of [35] is concurrent with [18] and [1].

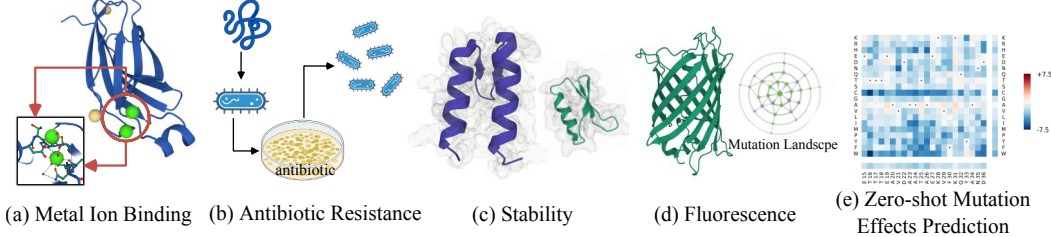

(a) Metal Ion Binding    (b) Antibiotic Resistance    (c) Stability    (d) Fluorescence    (e) Zero-shot Mutation Effects Prediction

Figure 2: Protein function prediction tasks. (a) and (b) are annotation prediction tasks. (c)(d) and (e) are fitness prediction tasks.

## 3 Preliminaries

### 3.1 Tasks and datasets

We evaluate models on two structure prediction tasks:

1. Secondary structure (SS): This is a residue-level sequence-to-sequence task where each residue $x_i$ of a protein sequence $x = \{x_1, x_2, ..., x_L\}$ is mapped to a label $y_i$ corresponding to one of eight secondary structure tasks $y_i \in \{G, H, ..., C\}$ [28]. SS prediction examines the degree to which a PLM learns local structure.

2. Contacts: For a given protein structure, two residues are considered to be in contact if their $C_\beta$ carbons are within 8Å. We evaluate on pairs that are more than 6 positions apart in the primary structure [28]. We measure the results using Precision@$L$, which stands for the precision for the top-$L$ pairs with the highest predicted contact probability [47]. $L$ is the length of the protein sequence.

For both contacts and secondary structure, we use the dataset in [35] which is constructed from SCOPe [15], and use the suggested split as the training and testing sets (see Table 2). One concern is that the dataset used here has been trained by AlphaFold as they come from the Protein Data Bank (PDB) [3]. Hence, we investigate 48 additional proteins, which were collected from CAMEO[†] (Continuous Automated Model EvaluatiOn) with 'hard' category from 2021-08-28 to 2022-04-30.

We also evaluate on two function (annotation) classification tasks:

1. Metal ion binding (MIB): This is a binary classification task, where a PLM with a new classification layer is used to determine whether there are metal ion–binding sites in the protein. The dataset is also collected from PDB with annotation as metal ion binding. We randomly sample the same amount of proteins from the database as the negative class.

2. Antibiotic resistance (ABR): This is a multi-class classification task, where a PLM need to correctly determine which class of antibiotic a protein degrades. We construct the dataset from CARD [26] which contains 19 different antibiotic types (see Appendix A.5 for details).

And three fitness prediction tasks. Unlike functional annotation prediction, protein sequences in this task are all from the same wild-type with a small number of mutated residues.

1. Stability: This is a protein-level regression task that predicts the protease concentration at which a protein can maintain its fold [36]. We use the data splits from TAPE.

2. Fluorescence: This is also a protein-level regression task, predicting the log-fluorescence intensity of the protein sequence [37]. We use the data splits from TAPE.

3. Zero-shot mutation effect prediction: This is a protein-level prediction task by comparing the difference between likelihoods assigned to the mutated residue and the likelihoods assigned to the wild-type (see [27] for details). We evaluate five protein mutation datasets from DeepSequence [34]. Only single point mutation data is considered in this sub-task.

---

[†]https://www.cameo3d.org/

Table 2: Dataset descriptions

| Task | Source | Train | Test |
|------|--------|-------|------|
| Secondary Structure & Contact Prediction | SCOPe | 11680 | 3617 |
| Metal Ion Binding | PDB | 6000 | 1332 |
| Antibiotic Resistance | CARD | 2072 | 1344 |
| Fluorescence | TAPE | 21446 | 27217 |
| Stability | TAPE | 53614 | 12851 |

## 3.2 Methods

For supervised function prediction tasks, we remove the original classification layer for ESM-1b and MSA-Transformer, and the Structure module for AlphaFold. These remaining parts are called protein representations. We then add a linear layer on top of these representation models to perform new classification or regression tasks. We adopt the standard fine-tuning strategy by fine-tuning all parameters using Adamw optimizer with 1e-5 as learning rate. For the zero-shot fitness prediction task, no training is needed. Instead, we obtain the softmax output at the corresponding mutation site in each protein sequence. The probability value is regarded as the fitness value for each amino acid following [19, 27]. As for structure predictions, we follow the same practice in [28]. Other details can be seen in Appendix A.1.

# 4 Results

## 4.1 Structure Prediction

In this section, we examine the structure representation ability of the three PLMs models discussed above. In general, the secondary structure and contact prediction tasks do not have significant practical values since there is already highly accurate 3D structure data by AlphaFold. The purpose instead is to provide a reference for function prediction, given that some tasks have similar formulation.

Table 3 and 9 (Appendix A.4) show the results of SS prediction and contact prediction, respectively. For SS prediction, we consider two settings: pre-trained parameters and training from scratch (i.e., with random initialization). For Evoformer, pre-training means the training of AlphaFold. For contact prediction, all PLMs have been already pre-trained since there is no meaningful contact map without pre-training.

First, we can easily observe that with pre-training, Evoformer performs the best in both tasks. Particularly, Evoformer outperforms ESM-1b and MSA-Transformer by a large margin for contact prediction with over 94% accuracy on Precision@$L$ (see Table 9). By contrast, it does not perform as accurately as the contact prediction task for SS prediction, which improves ESM-1b by 11.6% and MSA-Transformer by 5%. This is reasonable since SS prediction is different from 3D structure prediction from the machine learning perspective. Even though Evoformer has strong structure representation ability, it may not work well by just adding a linear classification layer. This suggests that a more complex structure module for SS prediction is necessary for higher accuracy. In comparison, since contact information can be directly extracted from the pairwise distance map, it is not surprising that Evoformer shows superb results in this task. Thereby, we have also demonstrated results of 48 new proteins in Figure 3, where these proteins have not been trained by AlphaFold. In general, they are consistent with our above analysis.

Second, we observe that pre-training is important: all PLMs are remarkably improved with around 40% improvement for ESM-1b and 28% for Evoformer. Interestingly, we note that Evoformer performs worse than MSA-Transformer by training from scratch. We conjecture that Evoformer is much harder to be trained without good initialization as its network architecture is more complex and deeper than MSA-Transformer.

Finally, for structure prediction, evolution-aware models are largely better than evolution-free ESM-1b. This is consistent with biological intuition and previous observation, such as in [30].

Table 3: SS prediction. 'Scratch' means training from scratch without pre-training.

| Model | Pre-train | Scratch | Improv. |
|---|---|---|---|
| ESM-1b | 0.703 | 0.500 | 40.6% |
| MSA-Transformer | 0.748 | 0.634 | 18.0% |
| Evoformer | **0.785** | 0.614 | 27.8% |

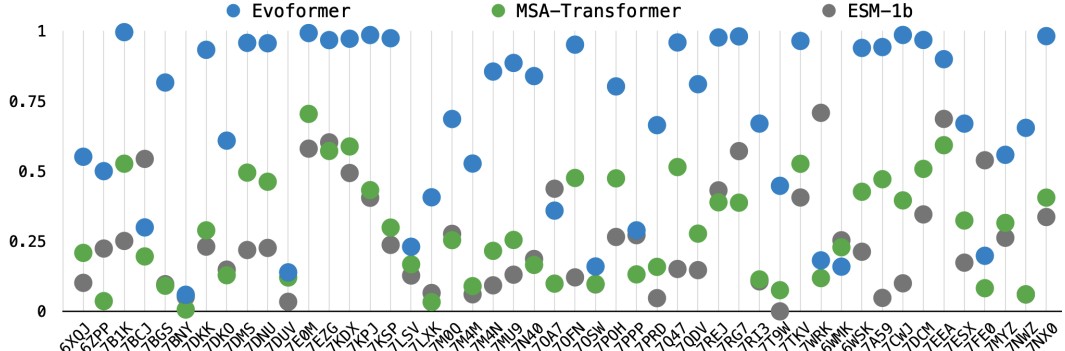

Figure 3: Contact map prediction with the CAMEO dataset (from 2021-08-28 to 2022-04-30) in terms of Precision@$L$. These proteins have not been used for training AlphaFold.

## 4.2 Supervised Function Prediction

AlphaFold predicts very accurate protein structures. However, it remains unknown whether its core representation module Evoformer can prediction function. Similarly, we investigate MSA-Transformer's ability to predict protein function, which is unknown either. Note that, unlike annotation prediction, evolution-aware Evoformer and MSA-Transformer only take single sequences (rather than MSAs) as input since all protein variants share the same MSAs for the fitness prediction task.

The results of annotation prediction (protein-level classification) and fitness predictions (protein-level regression) are listed in Table 4 and 5 respectively. We use classification accuracy [28] for evaluating annotation prediction and Spearman rank correlation [27] for evaluating fitness prediction. First, we find that for annotation prediction, ESM-1b, the evolution-free PLM, exceeds the two evolution-aware PLMs, although opposite observations are made for predicting structures. Specifically, pre-trained ESM-1b achieves 5.8% and 17.4% improvements over pre-trained Evoformer and MSA-Transformer. This is a bit surprising since the biological intuition is that protein functional properties are mediated by structures. An important conclusion we reached here is that **better structure PLMs do not mean they have a better representation for predicting function.**

Second, all pre-trained PLMs perform very well on the ABR task, a multi-class protein classification task. Despite that, we can still observe an obvious improvement between training-from-scratch and pre-training. Similar observation brought from pre-training can be obviously seen from all other function prediction tasks. This suggests that **both supervised and unsupervised pre-training on PLMs are very useful to obtain protein function representations.**

Third, we observe that the three pre-trained PLMs in general show better results than the one-hot [28] and ResNet [28, 17] baselines in Table 5. This observation is new since neither Evoformer nor MSA-Transformer have been investigated for supervised fitness prediction. In more detail, ESM-1b performs the best on the fluorescence task, whereas Evoformer is the best on the stability task. This may be because protein stability has a closer relationship with protein structure, or because sequence-only pretraining does not generalize as well as structural pretraining to *de novo* miniproteins.

## 4.3 Zero-shot Mutation effects Prediction

Recent work revealed that PLMs are strong zero-shot learners in predicting potential viral mutations [19, 20]. We are interested in evaluating Evoformer's masked-residue reconstruction ability in such a zero-shot setting given that it also has a BERT-like masked token prediction loss.

Table 4: Functional annotation prediction. 'scratch' means random initialization for parameters.

| Model | Pre-train | | Scratch | |
|---|---|---|---|---|
| | MIB | ABR | MIB | ABR |
| ESM-1b | **0.840** | **0.979** | 0.628 | 0.945 |
| MSA-Transformer | 0.715 | 0.961 | 0.640 | 0.932 |
| Evoformer | 0.794 | 0.979 | 0.645 | 0.920 |

Table 5: Fitness prediction. Scores are |Spearman $\rho$| on each task.

| Model | Pre-train | | Scratch | |
|---|---|---|---|---|
| | Fluorescence | Stability | Fluorescence | Stability |
| One-hot [28] | 0.14 | 0.19 | - | - |
| ResNet [28] | 0.21 | 0.73 | 0.28 | 0.61 |
| ESM-1b | **0.68** | 0.76 | 0.68 | 0.59 |
| MSA-Transformer | 0.64 | 0.67 | 0.67 | 0.61 |
| Evoformer | 0.67 | **0.81** | 0.36 | 0.52 |

Figure 4 shows the results of all PLMs. First, we observe that ESM-1b and ESM-1v [27] yield the best results, and both of them are evolution-free models. ESM-1v shares an architecture with ESM-1b, but is an ensemble of 5 models trained on UniRef90 instead of UniRef50. MSA-Transformer in general performs well except on BLAT_ECOLX_Ostermeier2014. We notice that MSA-Transformer is highly affected by the quality and sequence identity thresholds of MSAs. For our experiments, we use the default setting by HHblits [33] searching from BFD[‡] database (see Appendix A.3).

The most surprising results are for Evoformer, which performs poorly in this task on all five datasets. To examine this cause, we present results of another three models, namely ESM1-85M (released by [35]), ESM1b-88M (af2_data) and MSA-Transformer (af2_data) (see model descriptions in Figure 4). As shown, ESM1-85M performs much better than ESM1b-88M, which indicates that the accuracy of the zero-shot fitness prediction task is largely affected by the size of the pre-training dataset. This may explain why Evoformer performs so poorly in this task. Beyond this, there is another key difference between these models. Evoformer or AlphaFold were trained with five different loss functions, among which four are structure-related loss and only one auxiliary loss is the masked token reconstruction (MTR) loss. By contrast, MSA-Transformer and ESM-1b were trained with only the MTR loss. This may also be a reason for explaining the worse results of Evoformer.

**Answer for Q(i): Yes**. The answer is provided in Table 4 and 5. From these results, we can see that Evoformer with pre-trained parameters largely outperforms the non-pretrained version in four different function prediction tasks. Together with its structure representation ability, we conclude that **parameters learned by AlphaFold are general-purpose and useful to various structure and function prediction tasks.**

**Answer for Q(ii): No**. The answer is provided in Table 3, 9, 4, 5 and Figure 3. These results show that Evoformer indeed outperforms MSA-Transformer and ESM-1b when predicting structures, but it does not obviously outperform ESM-1b when predicting function. In particular, it does not work at all in the zero-shot mutation effect prediction task. Accordingly, we can conclude that **the AlphaFold-triggered revolution for structure prediction cannot be directly transferred to function predictions. ESM-1b is still the SOTA when predicting protein functions.**

## 4.4 Effect of MSA

Here, we want to understand the influence of MSAs on evolution-aware PLMs. Further, we explore the relationship between evolution-aware and -free models. As mentioned in Section 4.2, evolution-aware PLMs take only a single sequence as input for the supervised fitness prediction tasks, so we do not consider the affect of MSAs for these two tasks.

---

[‡]https://bfd.mmseqs.com/

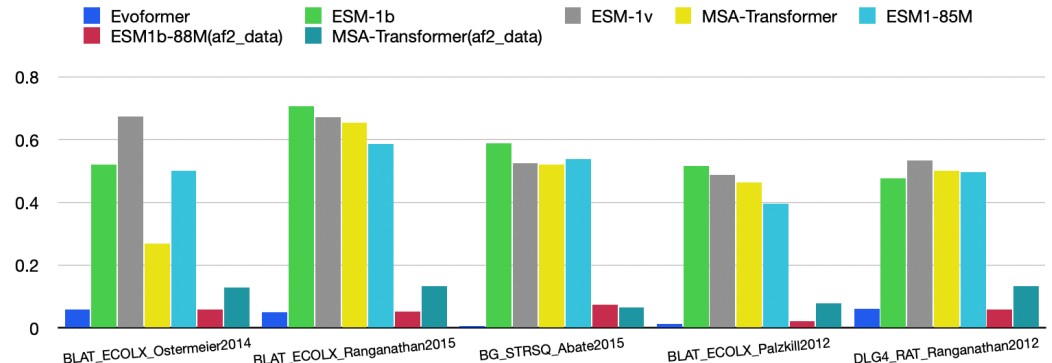

Figure 4: Zero-shot fitness prediction. ESM-1v's results is the average performance of five models; ESM1-85M was trained with the same dataset as ESM-1b but with a much smaller model size (85 million parameters); ESM1b-88M(af2_data) and MSA-Transformer(af2_data) were trained with the same protein dataset as used for training AlphaFold (including both the PDB data and those from Uniclust30 used in the self-distillation process) and have similar model sizes as Evoformer.

Table 6: Impact of MSAs. 'Seq' denotes an individual sequence, i.e., no MSAs.

| Model | Pretrained | SS | | MIB | | ABR | |
|---|---|---|---|---|---|---|---|
| | | MSA | Seq | MSA | Seq | MSA | Seq |
| Evoformer | Yes | 0.785 | 0.716 | 0.794 | 0.724 | 0.979 | 0.983 |
| MSA-Transformer | Yes | 0.748 | 0.631 | 0.715 | 0.707 | 0.961 | 0.908 |
| Evoformer | No | 0.614 | 0.624 | 0.645 | 0.632 | 0.920 | 0.875 |
| MSA-Transformer | No | 0.634 | 0.526 | 0.640 | 0.579 | 0.932 | 0.909 |

We report results on Table 6 and Figure 5. The MSA setting in Table 6 is consistent for both model training and inference. For Figure 5, we only consider the inference stage because it is a zero-shot task. Clearly, we observe that without MSAs, Evoformer, and MSA-Transformer both yield much worse results (with or without pre-training). Similarly, as shown in Figure 5, MSA-Transformer is also affected by the number of MSAs. In general, it performs much worse with few MSAs. These results show that (**Answer for Q(iii)**) **MSA data is a crucial input for evolution-aware PLMs, for not only the structure prediction but also the function prediction tasks.**

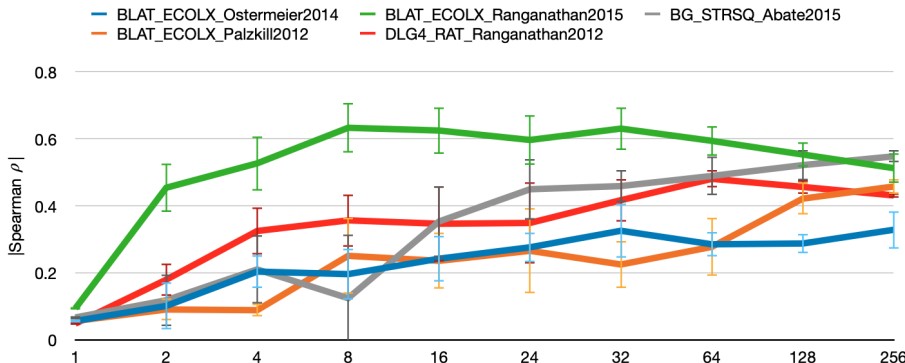

Figure 5: The effect of MSA depth on the zero-shot fitness prediction task using MSA-Transformer. Error bars are the standard deviation of six replicate experiments with randomly sampled $k$ MSAs.

### 4.5 Relationship between Evolution-aware & -free PLMs

Evolution-aware PLMs require MSA data as input for both training and inference. However, database search using HHblits [33] or Jackhmmer [22] can take up to 10 minutes for a 1000-residue protein. On the other hand, given the powerful function representation of ESM-1b, we design a fast homology retrieval framework using Siamese ESM-1b, following [31]. Figure 6 is the schematic of the ESM-1b-constructed MSA generating and serving process, called ESM-MSA (see Appendix A.2).

Table 7: MSA Impacts. 'Seq' denotes an individual sequence i.e. no MSAs.

| Model | SS Prediction | | | MIB | | |
|---|---|---|---|---|---|---|
| | Jackhmmer | ESM-MSA | Seq | Jackhmmer | ESM-MSA | Seq |
| Evoformer | 0.785 | 0.776 | 0.716 | 0.794 | 0.766 | 0.724 |
| MSA-Transformer | 0.748 | 0.733 | 0.631 | 0.715 | 0.774 | 0.707 |

Table 7 shows the results by using different MSAs for Evoformer and MSA-Transformer. MSAs generated by ESM-1b are comparable to those found by Jackhmmer (or HHblits) while being much faster (see Appendix A.3.2), suggesting that (**Answer for Q(iii)**) **the function representation learned by evolution-free ESM-1b can be used to construct MSAs for evolution-aware MSA-Transformer and Evoformer.**

## 5 Conclusions and Limitations

In the paper, we have presented empirical studies to characterize three successful PLMs. In particular, we focus on the function representation capacity of Evoformer. We draw several important conclusions: (i) Evoformer encodes not only structure, but also various protein functional properties; it provides an alternative choice when predicting protein functions, especially the stability prediction task; (ii) despite that AlphaFold dominates the structure prediction tasks, Evoformer is not yet able to substitute ESM-1b and MSA-Transformer when predicting protein functions; (iii) the functional predictability of evolution-aware PLMs is also influenced by protein MSAs. Interestingly, evolution-free ESM-1b can generate sufficiently-accurate MSAs for evolution-aware PLMs with much higher efficiency. Our work points out both strengths and weakness of three SOTA PLMs, which are potentially useful to many downstream biological tasks, including but not limited to protein engineering [8], cancer early detection [16], and drug discovery [24].

An important limitation is that we probe the representation capacity of these PLMs by only adding a linear head on top — a common practice in the deep learning community. While this verifies our findings, we believe such a linear layer is not sufficiently expressive to achieve SOTA results. That is the reason why Evoformer with a eight-layer Structure module achieves much higher 3D structure accuracy but with a linear layer it achieves less than 80% classification accuracy for the secondary structure prediction task. In addition, we do not fully disentangle the effects of the Evoformer architecture, training data, and loss function when comparing it to MSA-Transformer. Therefore, an important direction is to develop more advanced functional modules for higher prediction accuracy.

## Acknowledgement

This work is supported by the National Key Research and Development Program of China (No. 2022ZD0115100), the National Natural Science Foundation of China (No. U21A20427), the Westlake Center of Synthetic Biology and Integrated Bioengineering (WE-SynBio), the Research Center for Industries of the Future (No. WU2022C030), and Key Research Project of Zhejiang Lab (No. 2022PG0AC02).

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
