# OpenReview forum: "Exploring evolution-aware & -free protein language models as protein function predictors"
_NeurIPS.cc/2022/Conference — NeurIPS 2022 Accept_

### Official Review · Reviewer_ZqSX · 2022-06-22

**Rating:** 4
**Confidence:** 1
**Soundness:** 2 fair
**Presentation:** 4 excellent
**Contribution:** 2 fair

**Summary:**

The authors divide the sequence alignment representation module into two types: revolution-based and revolution-free, and examine the usefulness of the evlutionary information not only in the framework of protein structure prediction but also other prediction problems, related with proteins. The "revolution-based" approach is confined to "Evoformer" in this paper.


**Questions:**

-- Seeing the three models (a, b and c) in Fig. 1, can the models be compared in more detail? For example, can the authors consider more valiation of using the information?
-- Also how much can the performance be affected by changing the values of hyperparameters?


**Limitations:**

-- This work is significant from a ablation study viewpoint in bioinformatics. We might see what information can contribute to improve protein-related bionformatic problems, such as secondary structure or function prediction. However, methodological contribution would be definintely limited. One better presentation might be to do experiments more: for the same problem and same module, different ways of using the module are examined and the best way of using that is shown. However, I am unsure if this way could be good enough to be accepted for publicaiton by this conference, though.

**Strengths And Weaknesses:**

Strengths
-- The paper addresses the problem of examining if evoluationry information (or Evoformer) is more useful / important than multiple sequence alignment without evolutionary information. This problem would be important and also attractive from a bioinformatics (or biology) viewpoint.
-- Rather clear conclustions were lead by empirical studies. Particularly, it is indeed interesting that Evoformer does not necessarily work well for function prediction, comparing with

Weaknesses
-- Although the information is important, the problem might be minor.
-- The work is totally empirical. That is, the conclusions are obtained totally empirically. So although conclusions might be useful, the result of this work might not be universal. I think this is one of the points already mentioned by the authors in Conclusion.

---

> ### Author Response · Authors · 2022-07-31
> **First, thank you for the comments. We answer your questions as below.**
>
> Q1: The work is totally empirical. That is, the conclusions are obtained totally empirically
>
> Our paper is an empirical study work, however, the studied three protein language models (in particular AlphaFold’s Evoformer) have very high impacts in the community, they are probably by far the most powerful, nearly the largest, and most widely used (in biology) and well-known (to biologists) protein models. The focus of this paper is not to proprose a new neural network architecture or a new methodology, instead, we want to reveal the characteristics of these popular large-scale protein language models  for both AI and biology the communities.
>
> Understanding Evoformer or AlphaFold (landmark work) is of great importance and helpful for a large body of biologists who work on protein structures and functions (one of the most important research topics). In this paper, we have made many insightful observations that are unknown for the community.  For example, we found that finetuning Evoformer can be used to predict protein stability, which is a key weakness for AlphaFold (it can only predict accurate 3D structures for the foldable and stable proteins).  More importantly, Evoformer performs the best on this task, which is very important for protein engineering and drug design. We found that Evoformer or AlphaFold fails in the zero-shot prediction task, which are important cons compared with MSA transformer and ESM1b. These mutation effect prediction task are very very important for a variety of biological applications. Due to limited paper space, compute resources, and time, we cannot report too many tasks, but the reported results in the paper are very representative, for example fluorescence and stability for protein engineering, metal ion binding and antibiotic resistance for Microbiome and zero-shot prediction for virus and cancer mutation effect prediction.
>
>
> Q2: One better presentation might be to do experiments more
>
> Honestly, training Evoformer/AlphaFold is very challenging even by finetuning. That is the reason by far there is no even one paper that investigates the features of Evoformer. Only very few groups such as Deepmind (or Facebook) themselves did some improvements or training of AlphaFold. Almost all existing work only use AlphaFold as a black box due to its complex mechanism. Training it is super slow and requires very large memory GPUs. Some evaluations for the function prediction tasks could take half a month or even longer time for training on advanced GPUs(e.g., 8 A40s with 48G GPU memory), excluding the MSA preparation time (another two weeks with 100CPU cores). Each task we need to perform many different evaluations under different settings, e.g. with/without MSA, with/without pre-training. We have also evaluated many fine-tuning strategies, and many different heads (not only linear head used in the paper, e.g. pooling, attention, and more DNN layers). Since these strategies did not obtain some obvious improvements, we did not report them in the submission. To this end, we ran over 400 experiments for more than 30,000 GPU hours (running 8 A40s for about half a year for only experiments) in total to evaluate these three super large models on all tasks and datasets.
>
> Q3:  Also how much can the performance be affected by changing the values of hyperparameters?
>
> We use the pre-trained models of the three protein language models. Most hyper-parameter searching happened for pre-training, which was already done by other researchers. There are only a few hyper-parameters needed to be tuned, such as learning rate, batch size and regularization. We performed very basic searching, e.g., changing from 0.00001 to 0.001 with a fixed step size, e.g., {0.00001, 0.00005, 0.0001…}. Batch size is very small since these models are too large to fit the GPU memory. Here we can only set it to 1 for one 48G GPU card for Evoformer, and run the experiments using 8 cards. MSA-transformer can be set to 1 or 2 per card. ESM1b can be set to 1 or 2 or 4 (for shorter sequences). In general, we found that these finetuning models are relatively insensitive to these hyper-parameters with sufficient training data used in this paper. Task such as mutational effect prediction does not need choose hyper-parameters because it is zero-shot prediction.

---

### Official Review · Reviewer_RhwA · 2022-06-23

**Rating:** 6
**Confidence:** 5
**Soundness:** 2 fair
**Presentation:** 3 good
**Contribution:** 3 good

**Summary:**

The paper studies the effectiveness of evolution-based neural architectures including  Evoformer, MSA-Transformer, and evolution-free methods like ESM in protein structure and property prediction problems. Evoformer is similar to MSA-Transformer which learns protein representation from MSA, but Evoformer was used as part of the Alphafold for 3D representation prediction. While ESM is pretrained using only sequence information without the need for MSAs.
The authors examined these models in different standard structure predictions and protein properties prediction tasks. Their conclusion was that evolution-based neural architectures including  Evoformer, and MSA-Transformer outperform evolution-free methods like ESM in structure prediction tasks while Evoformer pretrained with much smaller data is bad at properties prediction tasks and zero-shot mutation prediction tasks.
The authors also study the effect of varying MSA quality on the results of the evolution-based neural architectures and proposed an approach that use ESM  for constructing ESM-MSA that is much faster than the standard MSA search but still achieves comparable results to the methods that use MSAs.

**Questions:**

For completeness, I think the experiments should include all benchmark data in TAPE, for example, homology prediction is missing.
Why don't you use the TAPE train/dev/split for contact map and ss prediction instead of SCOPe?
Table 2: missing information about dev set.


**Limitations:**

YES

**Strengths And Weaknesses:**

Strengths
This is a nice work that performs very careful empirical studies, well-written and accompanied with source code and two new standard benchmark datasets, this is a plus as the standard 5 benchmark datasets in TAPE are not enough for evaluating recent development in the field.

All methods were compared using the same linear layers for downstream tasks, even the authors see this is as a weakness as stated in the conclusion because for different downstream tasks better than linear layers should be considered to achieve the SOTA results but for comparisons of pretrained models, it is fair to use the same linear layers for all the tasks.

Some of the findings are interesting, for example, the results of Evolformer on the zero-shot mutations.

Weaknesses

It is important to note that Evoformer, MSA-Transformer, and ESM-1b are very different in network size, and different training data used. It is very hard to compare these pre-trained models and draw some conclusions on whether one method is better than another. I see that the authors have mentioned this issue in the introduction section and I agree that it is useful to check which pretrained models are better because most companies and researchers from academia do not have the computing resource to pretrain these large models. Yet the answer to that question only has a practical meaning which might help industries choose the right model for their application but not give an answer to the question of which methods are better (assuming they are the same size and pretrained on the same data).

This is an empirical study so, besides interesting benchmark results, its technical and novelty contribution are limited.
I evaluated the paper based on the technical depth, novelty contribution, and usefulness of the research to the community.

---

> ### Author Response · Authors · 2022-07-31
> **Thank you for the thorough review and the kind words regarding readability and reproducibility and research value.**
>
> Q1； This is an empirical study so, besides interesting benchmark results, its technical and novelty contribution are limited.
>
> Yes. This paper is mainly an empirical study on three most impactful protein language models in the community. Roughly speaking, the three models are by far the most powerful, (nearly) the largest, and most widely used (in biology) and well-known (to biologists) protein language models. All of them are trained by research teams in industry, i.e. Facebook and DeepMind, with several hundred GPUs/TPUs cards and tens of thousands CPU cores for searching MSAs. In particular, AlphaFold is super important for both AI and biology fields, it has led to a revolution. However, until now, very very few research work focus on deciphering AlphaFold rather than treating it as a black box. We are the first to study Evoformer (besides AlphaFold), the key module of AlphaFold. and a new types of protein language model.
>
> Q2；One concern is that these three models have different training data and model sizes, so it is not very convincing to reach some fair conclusion.
>
> First, it is indeed almost impossible to compare these super large models under an extremely fair setting – e.g., training AlphaFold with the same model size (as ESM1b) probably need at least 500 advanced GPUs running for several months. More importantly, comparing them with the same data size is meaningless from the biology perspective because the number of experimental 3D structures is limited, about 100x smaller than that of protein sequences. AlphaFold by far has no chance to be trained using 100 million+ real protein structures (No such big dataset).
>
> But we argue that most observations we made are relatively fair. For example, even Alphafold is trained with less data and smaller model size, it still shows considerably better results in stability and structure prediction tasks. Even ESM1b is much larger than the other two Evolution-based models and with more training data, it is still much weaker for structure prediction tasks, which shows that evolution information is very important. Then, MSA data is not only useful for structure prediction tasks but also important for function prediction tasks, the conclusion of which does not rely on the same training data size or model size either. Evoformer is comparable in the Fluorescence prediction task even it has smaller model size. The most important conclusion is that all three large models benefit from pre-training. The pre-trained representation of Evoformer (with both supervised and self-supervised training loss) has proven to be effective for many structure and function prediction tasks but fail for the zero-shot task. In addition, we have also proposed a fast MSA construction module in the appendix with advanced deep learning techniques. In fact, most observations made in the paper does not need the same model size and training data.
>
> At last, this is the first work that performs function prediction evaluation on Evoformer, the key module in AlphaFold. A relatively fair evaluation should be helpful for the biology community, and potentially inspires more new cross-disciplinary work. In fact, it is also very challenging to perform these empirical studies. AlphaFold is a very complex system, and takes a lot of compute and requires almost the most advanced GPUs devices. Many evaluations for the function prediction tasks take half a month or even longer time running on the advanced GPUs (8 A40s with 48G GPU memory, batch size=1), excluding the MSA searching time (another 2 weeks with 100CPU cores). For each task we need to perform many different evaluations under different settings, e.g. with/without MSA, with/without pre-training etc.. Finally, we ran over 400 experiments for more than 30,000 GPU hours in total to evaluate these three super large models on all tasks and datasets
>
>
> Q3: Why don't you use the TAPE train/dev/split for contact map and ss prediction instead of SCOPe?
>
> Both TAPE and SCOPe are commonly used benchmark dataset.  SCOPe dataset has 8-class c while TAPE dataset has only 3-class for SSP task. We guess the 8-class classification may be a bit harder task. We choose to use SCOPe because it is also used in ESM1b literature so as for easier comparison. The original paper performs five-fold cross validation. We follow the same setting and compare it for the datasets without an explicit dev set. For the dataset used in TAPE, we use its dev sets.

---

> > ### Comment · Reviewer_RhwA · 2022-08-06
> > **Need more experiments with the right setting before drawing conclusions**
> >
> > Dear authors,
> > Thank you very much for providing detailed feedback. I have read all the interesting reviews. Below are my requests.
> >
> > I think that in an evaluation paper, it is important to perform a fair comparison so that useful conclusions can be made. Fair comparison means the experimental settings should be the same or at least very similar to each other.
> > In general, for this work, there are four aspects under consideration:
> > 1.  Network architecture (e.g. Evolformer vs BERT)
> > 2. Network size
> > 3. Training data type (e.g. MSA, 3D, sequences)
> > 4. Training data size
> >
> > All four aspects will contribute to the accuracy of the models. For example, for the zero-shot mutation effect prediction tasks or homology prediction tasks (missing in your paper), I believe that training data size is the most important factor. Imagine that the models need to remember different patterns of mutations and their effects, smaller training data should not be ideal because the models do not see enough mutation patterns, thus this might explain why Evoformer is not good at the zero-shot mutation effect prediction tasks.
> > In the paper, it is unfortunate that all models have very different sizes, and different architectures and were trained with different data types. It is not able to change these aspects, however, the authors can do their best with the most important aspect regarding training data size. Even though it is very difficult to scale Evoformer with larger training data (due to lacking computing resources in academia), the authors can re-train the ESM-1b and MSA-Transformer with smaller datasets. This is doable in academia as it is easy to scale these models to 200K sequences or MSAs, i.e. approximately the same size of the data used for training Evoformer.
> >
> > I think this new experiment, plus adding experiments on homology tasks will make the paper stronger and the conclusions more reliable.
> >
> > Currently, the conclusions were drawn while all 4 aspects are not the same in each method, and I see that the conclusions with different experimental settings in the paper are not reliable.
> > I have given a good score for this paper based on its clarity, open-source codes, and adding two new benchmark datasets. I would love to raise the score if the authors could provide additional experiments as I suggested.
> > Thank you and best regards,

---

> > > ### Author Response · Authors · 2022-08-09
> > > **Thanks for the opportunity to raise the score. The requested experiments have been added.**
> > >
> > > We have done the required experiments and release both source code（code/ESM-1b_Pretrain and code/Function/remote homology), and related datasets （MSA of remote homology detection is uploaded in Google Drive, original dataset is in TAPE paper）and related pre-trained model (https://drive.google.com/drive/folders/1iShEW8NcMIlWqxTRgsEaI_t5ahoHsixt?usp=sharing) in the appendix.
> > >
> > > (1) We added **new results in Figure 4 and explanation in Section 4.3 with red color**. We trained ESM1b (ESM-PDB-88M with 88M parameters, similar size as Evoformer) using the similar dataset as that of AlphaFold, including both the original PDB dataset and these used during self-distillation process, around 650K in total.
> > >
> > > We found the results of ESM-PDB-88M showed very bad results, similar as Evoformer. This confirmed our conjecture that the one key reason for Evoformer's poor results are simply because of insufficient training data. Please see our paper's detailed analysis.
> > >
> > > We did not add the results of MSA-Transformer since the current results can well reveal the performance degradation when there is insufficent training data. In addtion, searching MSA for 650K protein sequences requires at least 2 months with 500 CPU cores.
> > >
> > > (2) Regarding the result of remote homology detection (RHD) task, we have added it in the  **Appendix Table 10**. Please take a look. It shows similar performance behaviors as another two protein annotation tasks.
> > >
> > > RHD is a task kinds of between structure and function. It measures the capacity of a model in detecting structural similarity. However, the task is formalized as a protein-level annotation task, predicting which fold class the protein belongs to, or what kinds of functional shapes the protein has. In view of this, it can be categorized as either a structural task or a functional prediction task. In TAPE, the authors treated it as a separate task different from these typical structural prediction tasks (called an Evolutionary Understanding Task or Protein Annotation Task) since the typical structural prediction tasks （SSP, contact prediction and 3D structure prediction） are all about atom- or residue-level predictions, while RHD is a protein-level prediction, more close to the two annotation prediction tasks, very like the MIB and ABR tasks in this paper.

---

> > > > ### Comment · Reviewer_RhwA · 2022-08-09
> > > > **Still need MSA transformer experiments**
> > > >
> > > > Thank you for adding results of ESM with smaller training dataset. That is a great effort given short time. I therefore raise the score.
> > > >  Could you please add the final experiment related to MSA-transformer for 650K sequences? I think you still have time to perform this experiment until the camera ready if this paper get accepted.
> > > > If you follow this process: https://github.com/soedinglab/hh-suite/issues/212 you can scale-up  MSA search, in my experience, 500 CPUs should allow you to create MSA for 650K sequences  in 4-5 days.
> > > > So far I see that given similar data size, what matter most is the data type used for training. The conclusion will be more clear if we also have the result of the MSA-transformer.
> > > > best regards,

---

> > > > > ### Author Response · Authors · 2022-08-10
> > > > > **Thanks, we promise we will add these results of MSA transformer in our camera-ready**
> > > > >
> > > > > Dear reviewer,  we promise we would add results of MSA transformer as you suggested. Also thanks for provding us the the fast MSA searching guideline. Searching MSA is not a big problem, just takes a bit time and money for buying CPU services. Best Regards.

---

### Official Review · Reviewer_WXJh · 2022-07-09

**Rating:** 6
**Confidence:** 4
**Soundness:** 3 good
**Presentation:** 3 good
**Contribution:** 4 excellent

**Summary:**

This paper studies the representations of the AlphaFold Evoformer across protein structure and function prediction tasks and compares them to protein language models. The paper finds that while Evoformer produces state of the art predictions of structure, it falls behind protein language models on function prediction.

**Questions:**

The authors may consider more strongly highlighting the use of learned linear projection heads as this is critical for interpretation of the results; the authors may consider summarizing their findings in the abstract.

**Limitations:**

The authors have included a thoughtful discussion of the limitations of their work.

**Strengths And Weaknesses:**

EDIT: based on revisions by the authors during the discussion period and the addition of a held out test set for contact prediction I am raising my evaluation. However the concern about training set holdout also applies to the the other structure prediction result in the paper (secondary structure) and I suggest the authors address this using a similar held out set of structures whether it is accepted at Neurips or whether they resubmit to another venue.

This paper has several important contributions:
* Finding that protein language models perform better than AlphaFold on function prediction tasks is unexpected and interesting
* Studies the dependence of MSA based models on the additional sequences
* Investigates the use of MSAs produced by language model embeddings and finds that retrieval using embeddings is comparable to HMMs

Overall the findings and the investigations are of interest to the Neurips research community and I would very much like to recommend this paper for acceptance. However there is a major methodological concern that undermines some of the paper's main conclusions. In short the assessment of contact prediction is flawed as the test sets used are not held out for AlphaFold training. A number of conclusions in the paper depend on this comparison. Held out test sets, e.g. CASP14 and CAMEO proteins after the CASP14 cutoff date are available.

There are also some problems with the related work section. E.g. in the sentence "PLMs trained on these databases have been successfully applied for various protein related tasks" Rives et al 2021 didn't look at annotation prediction (the authors should cite Bileschi et al 2022 for this), but did look at secondary structure prediction, contact prediction, remote homology detection, and mutational effects. Heinzinger et al 2019. also looked at secondary structure prediction, as did Elnaggar et al 2020. Alley et al 2019, Meier et al 2021, and Hie et al 2021 also looked at fitness prediction. When discussing early work Rives et al 2021 and Heinzinger et al 2019 which were concurrent with Alley et al 2019 should be mentioned. The citation for ESM1b and recovery of contacts cites Vig et al. rather than Rives et al 2021 which introduced ESM1b and studied recovery of contacts. Rao et al 2021 and Vig et al 2021 followed this work up with discovery of the interpretability of attention maps as contacts and should also be cited for this.

---

> ### Author Response · Authors · 2022-07-31
> **We have added new results as requested**
>
> Q1: I would very much like to recommend this paper for acceptance. However, there is a major methodological concern that undermines some of the paper's main conclusions. In short the assessment of contact prediction is flawed as the test sets used are not held out for AlphaFold training.
>
> Thanks for this great question.In the original version, we indeed noticed this potential concern and thus provided six new proteins that are not trained by AlphaFold, as shown in Figure 3. We actually planned to report more proteins but we found the comparison results in Figure 3 are too obvious. We thought the community might not doubt the structure representation ability of Evoformer (its structure representation ability can be already well-established through AlphaFold).
>
> **Despite that, we believe it would be more convincing to evaluate more 'new' proteins, please take a look at Figure 3 of the newly updated version. We have present 48 latest proteins (with protein names) chosen from CAMEO (with category 'hard'), which are not used for training AlphaFold. We believe these results will be more convincing and hopefully address your main concern.**
>
> Q2: your revision opinion for related work
>
> **Thanks so much for pointing out this and we have updated our related work, please have a look marked with red color.**

---

> > ### Comment · Reviewer_WXJh · 2022-08-08
> > **Thanks and reply**
> >
> > Dear authors, thank you very much for the response and revisions.
> >
> > Q1. The inclusion of the CAMEO proteins is a significant improvement and addresses a major concern about the paper. However I disagree with the statement about secondary structure—it is trivially derived as a consequence of the 3d structure prediction task.
> >
> > Q2. The related work section as written remains inaccurate. E.g. is UniRep a biLSTM?
> >
> > Note regarding additional experiments with ESM-PDB-88M. Figure 4 and text implies this model was trained on PDB sequences? Although the AlphaFold model was trained on PDB in a first pass, the final model was trained with a much larger self distillation set on UniClust30.

---

> > > ### Author Response · Authors · 2022-08-08
> > > **Thanks for your insightful comments for revising our paper.**
> > >
> > > Dear reviewer, thanks for your revision again. We have carefully revised our paper based on your comments.
> > >
> > > There are three new questions and we hope to solve it.
> > >
> > > (1) "However I disagree with the statement about secondary structure—it is trivially derived as a consequence of the 3d structure prediction task."
> > >
> > > Thanks, we can revise the statement. We agree with you and knew that the secondary structure (SS) can be inferred from 3D structures. But from the machine learning perspective, they indeed belong to a different task with different objectives. Based on our results, It also shows that the 3D structure prediction accuracy cannot be perfectly transferred to that of the SSP task. Despite that, Evoformer is still the best among these models. If there is a better way to describe this, please let us know and we would revise it following your instruction.
> > >
> > > In fact, the main idea we want to express is that the structural representation of AlphaFold have been well recognized by the community. This can also be seen from the contact prediction results suggested by you.
> > >
> > > (2) The related work section as written remains inaccurate. E.g. is UniRep a biLSTM?
> > >
> > > We have revised the description.
> > >
> > > (3) Figure 4 and text implies this model was trained on PDB sequences? Although the AlphaFold model was trained on PDB in a first pass, the final model was trained with a much larger self distillation set on UniClust30.
> > >
> > > Thanks for pointing out this. We have added new results in Section 4.3 with red color.   Both the original protein data in PDB and these self-distillation set are considered to train the BERT or ESM model. Please see our paper marked with red color.

---

> > > ### Author Response · Authors · 2022-08-09
> > > **We have updated the results  by involving these proteins in the self-distillation stage.**
> > >
> > > Dear reviewer, we are very grateful for your valuable comments, which help us a lot for revising the  manuscript.
> > >
> > > 1. We noticed your comments mentioned that you were very much to recommend this paper for acceptance. The main concern is about the contact prediction task as the evaluation data was already trained by AlphaFold. As suggested, we have added new experiments and we found that you agreed that these new results largely improved the paper quality.
> > >
> > > 2. As for the related work, we have just revised it. Thanks again for your precious advice.
> > >
> > > 3. **I think the only issue now is the AlphaFold training set**. We thank you for pointing out this.
> > >
> > > We have added new experiments by training ESM-1b with both the original protein data in PDB and these 355K sequences from Uniclust30 that are used for the self-distillation of AlphaFold.
> > >
> > > Note the original PDB database contains about 770K chains but many of them are repetitive proteins and we have simply removed them. After the basic preprocessing, we achieved around 300K chains. The original AlphaFold paper did some further preprocessing by filtering sequences so as to rebalance the length distribution and remove higher identity sequences. Finally they achieved around 116K protein sequences in PDB. Here we simply use the 300K without too detailed pre-processing, which should be fine for training ESM1b.
> > >
> > > In total, we used about 300K+355K for training ESM1b, the results and analysis is shown in **Figure 4**. Please take a look.

---

> > > ### Author Response · Authors · 2022-08-09
> > > **Dear reviewer, we have revised the paper as you suggested, would you like to take a look.**
> > >
> > > Dear reviewers,
> > >
> > > We have revised our paper several times following your instructions, including adding new experiments about contact prediction, adding self-distillation set for training BERT;  revising the related work part. We have revised most places as you suggested.
> > >
> > > Would you like to give us an opportunity.
> > >
> > >
> > > Thanks and Best Regards!

---

### Official Review · Reviewer_TPVF · 2022-07-11

**Rating:** 8
**Confidence:** 5
**Soundness:** 4 excellent
**Presentation:** 4 excellent
**Contribution:** 4 excellent

**Summary:**

The paper investigates the PLM module in AlphaFold, Evoformer. In particular, it looks at assessing its representational ability beyond structure prediction, which is already well-established through AlphaFold. The paper does this by comparing the Evoformer module with two other popular PLMs: ESM-1b, MSA-Transformer in a number of tasks related to predicting protein functions. Particularly, the paper looks at:

- (1) Can the Evoformer module in AlphaFold produce representations helpful in predicting protein function?
- (2) Can Evoformer replace ESM-1b and MSA-Transformer for protein prediction functions?
- (3) How much do evolution-based PLMs rely on MSA input? Can evolution-free PLMs assist with MSA construction?

The paper compares metrics of models on a number of structure prediction, function classification and fitness prediction tasks. The paper concludes that:

- (1) Evoformer encodes protein functional properties, and can be used in predicting protein functions, in particular the stability prediction task;
- (2) Evoformer is not yet  able to substitute ESM-1b and MSA-Transformer models for predicting protein functions;
- (3) MSA data is crucial input for evolution-based PLMs, for both structure and function prediction tasks. Evolution-free ESM-1b can generate accurate MSAs for evolution-based PLMS.


**Questions:**

- Elaborating on the concern of using the SCOPe dataset (Lines 116-117)
- The low performance of Evoformer on the zero-shot fitness prediction task is interesting, particularly in comparison to the other models. The paper discusses this a bit in lines 215-220. It seems that the paper suggests that one potential reason for this is the difference in training between MSA-Transformer and ESM-1b and Evoformer. While not trained solely on masked token reconstruction, the masked token reconstruction objective is not missing from Evoformer. It would be helpful to speculate a bit more on why the difference would be so large here - is the paper suggesting that training on supervised protein structure tasks might degrade zero-shot performance on fitness prediction tasks?

Minor Miscellaneous Suggestions
- Line 157: ‘meaningful’, not ‘meanful’?
- Line 160: ‘By contrast, it does not perform as accurately as the contact prediction task for SS prediction’ -> I found the wording confusing; consider re-wording
- Line 189: ‘.. better structure PLMs do not mean they have a better representation for predicting function‘ -> also confusing, consider re-wording
- Table 5: Consider re-wording ‘Scratch’ to ‘Random Initialization’
- Consider Higher resolution for Figure 4



**Limitations:**

- Authors have adequately addressed the limitations of the work under 'Conclusion and Limitations'

**Strengths And Weaknesses:**

- This is a really well-written paper presenting interesting results that can be valuable for the field of PLMs and structural biology.
I believe the research questions being asked are valuable: assessing the representational power of Evoformer beyond structure prediction is interesting
- Follow-up work on the research questions started here could lead to insights around dynamics between structural and functional protein functions
- The paper is sound and has great presentation - is very clearly written and does a good job at outlining the research questions, experiments ran and clearly shows how the results support the conclusions that are drawn
- The results themselves are valuable: in particular; I find it interesting that Evoformer performed so poorly on the zero-shot fitness classification task, compared to ESM-1b and MSA-Transformer models and that Evoformer does not consistently outperform the ESM-1b and MSA-Transformer models of predicting protein function tasks, as one might expect, given its huge success on structure prediction tasks
- The conclusion drawn based on zero-shot fitness prediction tasks that ‘the AlphaFold-triggered revolution for structure prediction cannot be directly transferred to function predictions’ might be too strong based on the experiments ran. As mentioned under ‘Limitations’, it’s possible that more complex model architectures could lead to better results. I suggest addressing the low performance
- I appreciated the paper including their limitations, such as how more complex model architectures could lead to better results

---

> ### Author Response · Authors · 2022-07-31
> **Thinks for your positive comments and helpful feedback**
>
> Q1：The conclusion drawn based on zero-shot fitness prediction tasks that ‘the AlphaFold-triggered revolution for structure prediction cannot be directly transferred to function predictions’ might be too strong based on the experiments ran. As mentioned under ‘Limitations’, it’s possible that more complex model architectures could lead to better results. I suggest addressing the low performance.
>
> We agree. In this paper, we used the common setting, i.e. adding a linear head layer during finetuning of the PLM, which we also suspect that it might not be sufficiently enough. However, we had indeed tried some other intuitive ways in the beginning, e.g., adding 2 or 3 DNN layers, performing the basic pooling, or using an attention layer on top of Evoformer, exactly the same way as that used in FLIP [1], unfortunately, we did not find obvious accuracy gains compared with a very basic linear layer.  We believe that developing an advanced functional module as complex and effective as Structure Module in AlphaFold2 is interesting but also very challenging. Currently there has yet to be any related research. Hopefully, our findings and conclusions could be useful and instructive for the community and pave the way for new research to explore more advanced protein functional prediction module.
>
> Q2：Elaborating on the concern of using the SCOPe dataset (Lines 116-117)
>
> As mentioned in our paper, the proteins in the existing benchmark datasets (SCOPe and TAPE) are mainly from the PDB database, which might be already used for training AlphaFold. But we argue that this is fine for the SSP task --- although SSP belongs to a type of structure prediction task, it is a very different task from the 3D structure prediction from the machine learning optimization perspective, that is, the 3D structure prediction task is to generate all atom coordinates in the protein 3D space by the Structure Module (SM) of AlphaFold, while SSP is to generate the probabilities of the 8 secondary structure classes for each residue in the protein sequence by Evoformer; In this view, 3D structure prediction can be simply regarded as a pre-training task for SSP.
>
> The contact prediction task indeed has some correlation with 3D structure prediction task. Specifically, the contact prediction is to predict the distance of every two residues, while one objective of the 3D structure is to predict coordinates of all atoms in the protein. In fact, we notice this and simply reported results of six additional proteins that have never been trained by AlphaFold  (see Figure 3 of the submission version). We indeed planned to show more proteins in our submission version, but we found the comparison was very obvious and we also thought similarly as you mentioned in the comments, **the structure representation ability can be already well-established through AlphaFold** (it probably won't be very interesting to show many results about other low-level structure prediction tasks.).
>
> To make this part more convincing, we have updated our manuscript by **adding another 48 latest proteins from CAMEO, which are `new’ proteins and are not trained by AlphaFold**. Please refer to Figure 3 of the updated version （with red color）– exactly the observations can be made.
>
> Q3: It would be helpful to speculate a bit more on why the difference would be so large here - is the paper suggesting that training on supervised protein structure tasks might degrade zero-shot performance on fitness prediction tasks?
>
> **We have explained this issue by adding new results in Figure 4 and explanation in Section 4.3 with red color**.
>
> We trained ESM1b (ESM-PDB-88M with 88M parameters, similar size as Evoformer) using the similar training dataset as AlphaFold, including both the original PDB dataset and these used during self-distillation process, around 650K in total.
>
> We found the results of ESM-PDB-88M showed very bad results, similar as Evoformer. This confirmed our conjecture that one key reason for Evoformer's poor results are simply because of insufficient training data. Please see our paper's detailed analysis
>
>
> Other minor revision will be corrected in the camera-ready version.
> [1] Benchmark tasks in fitness landscape inference for proteins

---

### Meta-Review · Area_Chair_iea7 · 2022-08-26

**Recommendation:** Accept
**Confidence:** Certain

**Metareview:**

The paper provides an empirical study of the representation learning component of Alphafold for protein sequences, called Evoformer. The topic of protein language models (PLMs), of which Evoformer is an example, is relevant and interesting to the machine learning community. This is due to the success of Alphafold in protein structure prediction, as well as the popularity of large language models in natural language.

Four reviewers have carefully considered the results, and there was a very constructive interaction between the authors and three reviewers during the discussion phase. Thank you to both the authors and reviewers for engaging in good faith. All reviewers unanimously agreed that the paper is well written and the method reproducible (modulo compute). The revised paper provides useful empirical results to guide future researchers interested in representing protein sequences for different possible tasks.

It gives me great pleasure to recommend the paper for acceptance to NeurIPS 2022.

**Award:**

No

---

### Decision · Program_Chairs · 2022-09-14

Accept